

# Simplified support structure design for multi rotor wind turbine systems

Sven Störtenbecker[1], Peter Dalhoff[1], Mukunda Tamang[1], and Rudolf Anselm[1]

[1]Dep. Mechanical Engineering & Production, HAW Hamburg, Berliner Tor 21, D-20099 Hamburg, Germany

**Correspondence:** Sven Störtenbecker (sven.stoertenbecker@haw-hamburg.de)

**Abstract.** In this study different multi rotor wind turbine systems (MRSs) are designed in such a way that the space frame, forming the connection between rotor nacelle assemblies (RNAs) and tower, is modeled as an ideal truss work. To dimension the tube diameters and wall thicknesses, a simplified load case is used with an adjusted safety factor for loads. This simplified approach allows fast computations of a large variety of different support structure designs. By variation of rotor number, space
frame topology, space frame depth and the positioning of yaw bearings, it is possible to gain an understanding of the optimal MRS design. As such, the simplified approach is a preliminary step helping to choose a good design parameter combination for a more detailed and comprehensive analysis.

## 1    Introduction

In times of ever growing wind turbines and their components, the industry is facing new challenges in manufacture and trans-
portation, as well as loads and strength. A multi rotor wind turbine system (MRS) could overcome the obstacles of this growth trend.

Studies from the INNWIND project showed the potential of a 20 MW MRS with 45 rotors to reduce the levelized cost of energy (LCoE) compared to a power equivalent single rotor (SR) (Jamieson et al., 2017). Recent results from Vestas' four rotor MRS demonstrator revealed advantages in aerodynamic efficiency and in wake recovery when compared to a SR (van der Laan
et al., 2019).

Assuming a MRS with an overall capacity of 20 MW, due to the resulting lower rotor nacelle assembly (RNA) masses, based on the square cube law, and the load averaging effect (Jamieson et al., 2017), it should be more suitable to build up a MRS with a high number of small rotors, rather than a small number of large rotors. This can be categorized as a multi digit MRS (MD-MRS). To allow original equipment manufacturers (OEMs) to move towards MRSs by using their existing turbine
portfolio, a medium term solution might be the use of few rotors. This single digit MRS (SD-MRS) would be built up of three to nine rotors in the megawatt range. In the long term, a MRS with a high number of rotors using a newly developed small RNA in the kilowatt range seems favorable.

The DTU 10 MW research wind turbine (Bak et al., 2013) is used in this study as a basis for down- and upscaling. This includes downscaling to the size of the rotors used for the SD-MRS (set to 2 MW, 4 MW or 8 MW), downscaling to the size




of the MD-MRS rotors, as well as upscaling to a large SR reference wind turbine with an equal overall capacity. The MRS designs that have been analyzed are in the range of an overall capacity of 14 MW to 28 MW.

In this study different designs for the MRS are designed based on ultimate loads and buckling. The designs are built up in such a way, that the space frame is modeled as an ideal truss work. To dimension the tube diameters and wall thicknesses, a simplified load case of maximum thrust force at steady rated wind speed on all rotors and the gravitational forces resulting

from the RNA weights is used with an adjusted safety factor for loads. The axial forces of the truss members are calculated via finite element analysis (FEA). Diameter and thicknesses are first dimensioned against material strength and second if necessary against stability (Euler buckling). The following aspects in designing are not considered for reasons of simplicity: fatigue analysis of space frame and tower, local shell/plate buckling, dynamic behavior (modal analysis), design load cases according to (IEC 61400-1 Ed.4).

The weights of tower, space frame and RNAs are multiplied with cost per mass factors. The space frame topology is varied with respect to the depth of the structure and the yaw bearings positions. The optimum of each design is determined based on the minimal cost, the capital expenditure (CAPEX) for RNAs, space frame and tower. An assumed Rayleigh wind speed distribution is used for the annual energy yield for each design and results in very simplified levelized cost of energy (SCoE). SCoE means, that the operational expenditure (OPEX), the decommissioning expenditure (DECOMMEX), interest rate as well

as balance of plant (BoP) are not considered.

This simplified approach allows fast computations of a large variety of different support structure designs. By variation of rotor number, space frame topology, space frame depth and the positioning of yaw bearings, it is possible to gain an understanding of the optimal MRS design. As such, the simplified approach is a preliminary step helping to choose a good design parameter combination for a more detailed and comprehensive analysis.

## 45  2   Simplified support structure design

An overall capacity in the 20 MW range is assumed for the MRS. Regardless of the number of rotors, or rather the distinction of SD-MRS or MD-MRS, rotor data like the masses and diameters of rotors are needed. The basis for this rotor data is the DTU 10 MW research turbine (Bak et al., 2013) which is scaled down for the rotors of the MRS, as well as scaled up for a SR with an equal overall capacity.

Scaling is done under the assumption of similarity rules for wind turbines and a constant tip speed ratio (Jamieson, 2018). The rotor diameters for the scaled turbines $D_{\mathrm{rotor,i}}$ are calculated with the rotor diameter of the DTU rotor $D_{\mathrm{DTU}}$:

$$D_{\mathrm{rotor,i}} = \sqrt{\frac{P_{\mathrm{i}}}{P_{\mathrm{DTU}}}} \cdot D_{\mathrm{DTU}}. \tag{1}$$

The masses of rotors and nacelles are scaled via:

$$m_{\mathrm{i}} = \left(\frac{D_{\mathrm{rotor,i}}}{D_{\mathrm{DTU}}}\right)^n \cdot m_{\mathrm{DTU}}, \tag{2}$$

with the scaling exponent $n$. Due to the different influences on masses like new and lighter materials, better light weight design and higher experience in manufacturing, scaling is somewhat a critical task, especially for blades. Inter- and extrapolated





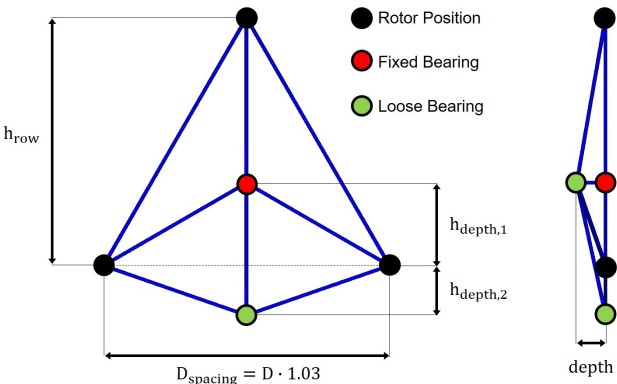

**Figure 1.** SD-MRS design no. 2, Tri rotor with bearing positions and design parameters

scaling trend lines also dependent on the considered data. In (Jamieson, 2018) this leads in one analysis to a scale exponent slightly above two. On the other hand, due to larger blade lengths, self weight bending could become a design driver, resulting in a higher exponent than three. Fundamentally blade mass is scaling in a cubic way.

The wind industry almost exclusively applies upscaling, due to the growth trend of turbines for a higher energy yield. For the MRS downscaling is of importance. Downscaling with an exponent around two seems not be suitable. Then the gain from new materials, technology and experience would be discarded and would result in a heavier and not modern blade.

An upscaling exponent of $n_{\mathrm{up}} = 2.6$ is set for blade and nacelle masses for the large SRs implying technological improvement. For downscaling an exponent of $n_{\mathrm{down}} = 3$ is set, assuming a scaled state of the art small turbine without any new future

improvements.

The simplified support structure design is described on the example of a SD-MRS with three rotors, as seen in Figure 1. An MRS support structure is composed of a tower and a space frame, connecting the RNAs among themselves and with the tower. The space frame consists of tubular steel connections.

In the INNWIND project (Jamieson et al., 2017) the spacing between rotors $D_{\mathrm{spacing}}$ was set to the rotor diameter plus five

percent:$D_{\mathrm{spacing}} = 1.05 \cdot D_{\mathrm{rotor}}$. Their simulations resulted in an increase of both thrust and power generation in comparison to a single rotor. A change to $D_{\mathrm{spacing}} = 1.025 \cdot D_{\mathrm{rotor}}$ resulted in no change to the mean value for thrust and power production. Here in this study a $D_{\mathrm{spacing}} = 1.03 \cdot D_{\mathrm{rotor}}$ is set.

All chosen layouts are based on equilateral triangles, so the vertical distance between rotor rows results to:

$$h_{\mathrm{row}} = D_{\mathrm{spacing}} \cdot \frac{\sqrt{3}}{2} . \tag{3}$$

The height of the first row results from a set blade tip ground clearance of $22\,\mathrm{m}$.

Tower and space frame are connected through yaw bearings, in this study always with a fixed/loose bearing combination. The MRS should be able to follow the wind via a global yaw system. Meaning that the whole space frame should be able to align itself perpendicular to the actual wind direction, rather than each rotor itself. How the yaw bearings would be connected





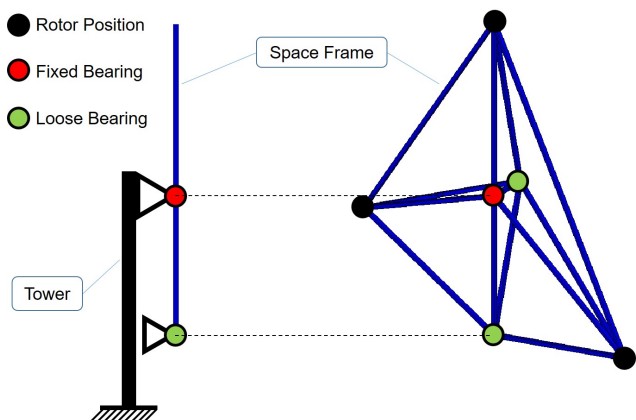

**Figure 2.** Tower and space frame connection via yaw bearings

to the tower in detail is of no importance for this preliminary concept design study. The gravitational forces caused by space
frame and RNAs are transferred through the fixed bearing to the tower. Thrust forces vertical to the rotor planes are transferred
through both, fixed and loose bearing.

  The vertical positions of the fixed and loose bearing, characterized with $h_{\mathrm{depth},1}$ and $h_{\mathrm{depth},2}$ is varied to investigate the
influence on the cost and to find the optimal design regarding the cost. The fixed bearing is in this study always on the tower
top, so the position also dictates the tower height, see Figure 2. In the designs with three rotor rows, the parameter $h_{\mathrm{depth},3}$ is
needed. Another geometric design parameter that is varied is the depth of the space frame.

  To dimension both tower and space frame a simplified design load case is defined: maximum thrust force (at steady rated
wind speed) on all rotors simultaneous. Because of wind shear, turbulent wind, gusts and the pitching behavior this is an
artificial and somehow unlikely case, but for this study it represents a worst case regarding the ultimate loads.

  The MRSs are designed in such a way that the space frame is modeled as an ideal truss work. To determine the member
forces of the space frame, a finite element analysis (FEA) carried out via ANSYS APDL is used. The space frame members are
modeled with bar elements. Bar elements have one local degree of freedom (DoF) per element node, the axial displacement,
resulting in three global DoFs per node. The corresponding reaction force to the local axial displacement is the local axial force.
The FEA requires initial diameters and wall thicknesses of the space frame members to determine and use the stiffness matrix.
Since the space frame is modeled as an ideal truss work the FEA solution of interest, the axial member force, is independent
of the initial cross section parameters.

  The use of bar elements implies that there are no other DoFs and reaction forces in the nodes, apart from the axial ones. In
reality the connections between the space frame elements and the rotors would be welded or bolted. Therefore shear forces as
well as bending and torsional moments would occur in the nodes. This could be modeled in the FEA via beam elements with six
DOFs and reaction forces per node. With the use of beam elements, the FEA solution would depend on the initial cross section
and the dimensioning process would be an iterative one for all elements/members and all space frame design configurations.
The differences between bar and beam element solutions was investigated and deemed neglectable for this preliminary study.



The thrust forces $F_\text{t}$ on the rotors are calculated via the $c_\text{T} = 0.827$ value of the DTU turbine at rated wind speed $v_\text{rated} = 11.4\,\frac{\text{m}}{\text{s}}$ (Bak et al., 2013), the scaled rotor diameter $D_\text{rotor}$ and $\rho_\text{air} = 1.225\,\frac{\text{kg}}{\text{m}^3}$ (IEC 61400-1 Ed.4):

$$F_\text{t} = \frac{1}{2} \cdot \rho_\text{air} \cdot c_\text{T} \cdot D_\text{rotor}^2 \cdot \frac{\pi}{4} \cdot v_\text{rated}^2 \cdot \tag{4}$$

This is still under the assumption of an unchanged tip speed ratio $\lambda$ due to scaling as well as unchanged $c_\text{P}$-$\lambda$ and $c_\text{T}$-$\lambda$ curves.

Also acting on the support structure are the gravitational forces of the RNAs with applied partial safety factor $\gamma_\text{f,gravity} = 1.35$. The partial safety factor for the thrust forces is set to $\gamma_\text{f,thrust} = 1.5$ instead to the suggested value of 1.35 according to (IEC 61400-1 Ed.4). This is due to the simplicity of the design and to compensate for the simplified load case. Both kind of forces are modeled as external forces acting on the rotor nodes of the space frame in the FEA.

The yaw bearings are modeled as boundary conditions with their respective DoFs. The fixed bearing disables all three spatial translations, the lower loose bearing has one DoF in the tower height direction. In Figure 2 a loose bearing is seen in the back behind the fixed bearing. This is required for the FEA simulation to run, otherwise the model wouldn't be kinematically determined. The space frame could still rotate and this would result in a singular reduced stiffness matrix.

The dead load of the space frame and the drag forces of both space frame and tower are neglected. This is due to the fact, 115   that both tower and space frame are going to be dimensioned and an iterative process is to be avoided.

The space frame members are first dimensioned against ultimate strength with applied safety factor for material $\gamma_\text{m} = 1.1$ and an assumption of a thin-walled tube: the wall thickness $t$ is much smaller than the diameter $D$: $t \ll D$.

A ratio for the wall thickness to diameter is defined $r_\text{t} = \frac{t}{D}$. This is set to $r_\text{t,b} = \frac{1}{120}$ for the space frame bars and $r_\text{t,t} = \frac{1}{250}$ for the tower. For the cross section follows:

$$A = \frac{\pi}{4} \cdot \left( D_\text{outer}^2 - D_\text{inner}^2 \right) \tag{5}$$

$$\approx \pi \cdot D \cdot t \tag{6}$$

$$\approx \pi \cdot D^2 \cdot r_\text{t}. \tag{7}$$

For both space frame and tower a construction steel with a yield strength of $\sigma_\text{yield} = 355\,\text{MPa}$ is used. Based on $\sigma = \frac{F}{A}$ and the FEA based axial forces $F_\text{bar,i}$, including both safety factors for loads, the bar diameters $D_\text{i}$ can be calculated now:

$$D_i = \sqrt{\frac{|F_\text{bar,i}| \cdot \gamma_\text{m}}{\sigma_\text{yield} \cdot \pi \cdot r_\text{t}}}. \tag{8}$$

If necessary, in case of a compression state in the member element, a redimensioning against stability (Euler's critical load) with applied safety factor for buckling $\gamma_\text{m,buckling} = 1.2$ is required. Since both ends of the members are free to rotate in theory, the column effective length factor $l_\text{k}$ is set to $l_\text{k} = l_\text{i}$, the whole length of each space frame bar element $i$. Euler's critical load is defined as:

$$N_\text{crit} = \pi^2 \cdot \frac{E \cdot I_\text{b}}{l_\text{k}^2}, \tag{9}$$





with Young's modulus $E = 2.1e5\,\mathrm{MPa}$ and area moment of inertia for bending $I_{\mathrm{b}}$. Again, like with the cross section $A$, a simplification for thin-walled bars can be used (Wriggers et al., 2007):

$$I_{\mathrm{b}} \approx \frac{\pi}{8} \cdot D^3 \cdot t \tag{10}$$

$$\approx \frac{\pi}{8} \cdot D^4 \cdot r_{\mathrm{t}}. \tag{11}$$

In case of a positive member axial force, the member is in a tension state and stability is of no concern. A negative axial force means a compression state. If the difference $N_{\mathrm{crit}} - F_{\mathrm{bar}}$ is negative, the bar diameter can be dimensioned with:

$$D_{\mathrm{i,buckling}} = \sqrt[4]{\frac{8 \cdot |F_{\mathrm{bar,i}}| \cdot \gamma_{\mathrm{m,buckling}} \cdot l_{\mathrm{i}}^2}{\pi^3 \cdot E \cdot r_{\mathrm{t}}}}. \tag{12}$$

The conclusive bar diameter is set to the maximum of $D_{\mathrm{i}}$ and $D_{\mathrm{i,buckling}}$.

The tower diameters and wall thicknesses are determined by the tower bending reaction moment. The bearing reaction
forces from the FEA solution are checked against the analytical solution. There, the space frame is assumed as a rigid beam supported through a fixed and a loose bearing, loaded with the thrust forces of the rotors. The reaction forces of the bearings are the external forces on the tower. These reaction forces induce tower bending reaction moments $M_{\mathrm{b}}$ in the tower. Based on the bending stress $\sigma_{\mathrm{b}} = \frac{M_{\mathrm{b}}}{W_{\mathrm{b}}}$ with the moment of resistance $W_{\mathrm{b}}$ the tower can be dimensioned. Again a simplification for thin-walled tubes can be used:

$$W_{\mathrm{b}} \approx \frac{\pi}{4} \cdot D^3 \cdot r_{\mathrm{t}}. \tag{13}$$

The tower diameter $D_{\mathrm{tower}}$ follows to:

$$D_{\mathrm{tower}} = \sqrt[3]{\frac{4 \cdot |M_{\mathrm{b}}| \cdot \gamma_{\mathrm{m}}}{\sigma_{\mathrm{yield}} \cdot \pi \cdot r_{\mathrm{t}}}}. \tag{14}$$

Since there are no bending moments at the tower top based on this simplified approach, the tower diameter would result in zero. The space frame is connected with the tower via a fixed bearing and therefore the gravitational forces resulting from the
RNAs and the already dimensioned space frame are acting as an axial force at the tower top. Similar to Equation 8 the tower top diameter is calculated. After dimensioning space frame and tower the volumes of each part can be calculated and in the next step the masses with the density $\rho_{\mathrm{steel}} = 7850\,\frac{\mathrm{kg}}{\mathrm{m}^3}$. Masses for RNAs, space frame and tower are now known and need to be multiplied with cost per mass factors. These factors are obtained from a turbine cost splitting (Jamieson, 2018), CAPEX assumption (Fraunhofer ISE, 2018) and selected free available turbine data. The resulting factors can be seen in Table 1. The
calculated cost factor values for tower, rotor and nacelle are in good accordance to (Jamieson et al., 2017). The assumption for the space frame cost factor is taken from their study.

Each SD-MRS design is simulated for each design parameter combination of $h_{\mathrm{depth,i}}$ and the depth of the space frame.

The energy yield of the MRS is determined based on a Rayleigh wind speed distribution with a mean wind speed $v_{\mathrm{mean}} = 10\,\frac{\mathrm{m}}{\mathrm{s}}$ (Wind turbine class I, (IEC 61400-1 Ed.4)) and reference height $h_{\mathrm{ref}} = 167.9\,\mathrm{m}$ (hub height of the INNWIND 20 MW single
rotor (Pontow et al., 2017)).





**Table 1.** Cost fractions after (Jamieson, 2018) and resulting cost per mass factors

|  | Cost fraction of turbine CAPEX [%] | Cost per mass [€/kg] |
|---|---|---|
| Tower | 21.9 | 2.5 |
| Space frame | - | 5 |
| Rotor | 29.7 | 16 |
| Nacelle | 48.4 | 18 |

With the cost and the annual energy production (AEP) of the designs the SCoE can be calculated:

$$\text{SCoE} = \frac{\sum \text{cost of RNAs, tower, space frame}}{n_l \cdot \text{AEP}} = \frac{\text{CAPEX}}{n_l \cdot \text{AEP}}, \tag{15}$$

with an assumed wind turbine lifetime of $n_l = 25$ years.

To compare and normalize the SCoE values, a power equivalent single rotor SR is designed with the same assumptions as
the SD-MRS (blade tip clearance, loads, etc.).

## 3   SD-MRS

To keep the design space somehow limited and to reflect currently available turbines on the market, the rotors for the SD-MRS
are set to a single capacity of 2 MW, 4 MW or 8 MW. Table 2 shows the downscaled DTU 10 MW rotor values for the 2 MW,
4 MW and 8 MW SD-MRS rotors. All layouts of the SD-MRS and the MD-MRS are designed with one set rotor capacity per
170  design, meaning that there is no mixture of rotor sizes in one MRS design.

**Table 2.** Scaled rotor values for the SD-MRSs and initial DTU rotor values

| $P$ [MW] | $D$ [m] | $m_{\text{rotor}}$ [t] | $m_{\text{nacelle}}$ [t] | $m_{\text{RNA}}$ [t] |
|---|---|---|---|---|
| 2 | 79.8 | 20.6 | 39.9 | 60.5 |
| 4 | 112.8 | 58.4 | 112.8 | 171.2 |
| 8 | 159.5 | 165.1 | 319.2 | 484.2 |
| 10 | 178.3 | 230.7 | 446.0 | 767.7 |

Possible rotor numbers are set to 3 (Tri rotor), 5 (Penta rotor), 7 (Hepta rotor) or 9 (Ennea rotor) rotors. An even numbered
SD-MRS would result in a cantilever design with or without steel ropes to reduce loads. These cantilever designs and therefore
even numbered SD-MRSs are not considered in this study. The uneven numbered SD-MRS layouts are all designed with
equilateral triangles which results in the highest packing density of the rotor area.



**Figure 3.** Overview and numbering of the SD-MRS designs

Three possible rotors and four possible number of rotors would result in 12 possible overall capacity combinations, ranging
from $3 \cdot 2\,\text{MW} = 6\,\text{MW}$ to $9 \cdot 8\,\text{MW} = 72\,\text{MW}$. The goal is to be in the 20 MW range, so five overall capacity combinations
ranging from 14 MW to 28 MW are set:

- Tri rotor: $3 \cdot 8\,\text{MW}$ resulting in 24 MW. The three rotors can be arranged in two ways: one rotor in the lower row and
  two in the upper row (increasing order, SD-MRS design no. 1) or in the upside down way (decreasing order, SD-MRS
  design no. 2). Design no. 1 has the advantage of a higher energy yield, based on wind shear, compared to design no. 2.
  The disadvantage of design no. 1 is the higher tower base moment, resulting in higher cost.
- Penta rotor: $5 \cdot 4\,\text{MW}$ resulting in 20 MW. The five rotors can also be arranged in an increasing or decreasing order.
  Two version are designed for increasing and decreasing order each, with the same rotor layout, but a slightly different
  arrangement of the bars.
- Hepta rotor: $7 \cdot 2\,\text{MW}$ resulting in 14 MW or $7 \cdot 4\,\text{MW}$ resulting in 28 MW. Additional to the increasing or decreasing
  order a circular/hexagonal arrangement is possible.
- Ennea rotor: $9 \cdot 2\,\text{MW}$ resulting in 18 MW. Two or three rows of rotors are possible, both with increasing and decreasing
  order.

This results in an overall number of 18 designs for the SD-MRS, as seen in Figure 3.

The design parameters shown earlier, $h_{\text{depth}}$ and depth are varied via unit less ratios and the cost are calculated. Since the
variation of these design parameters doesn't change the heights of the rotors, there is no influence of the variation on the energy
yield of each design. To find the minimum SCoE, the minimum cost (CAPEX) are determined.

Almost all designs have three design parameters, since they have two $h_{\text{depth}}$ parameters. In the case of the Tri rotor with
increasing number of rotors, design no. 1, there are only two design parameters and the cost based on the variations can be
visualized as a response surface, as seen in Figure 4. There the cost are normalized to the value of the 20 MW SR.

The edges in the response surface are due to changes in the load distribution and the compression/tension behavior of the
members, resulting in steep changes of the masses and therefore the cost.





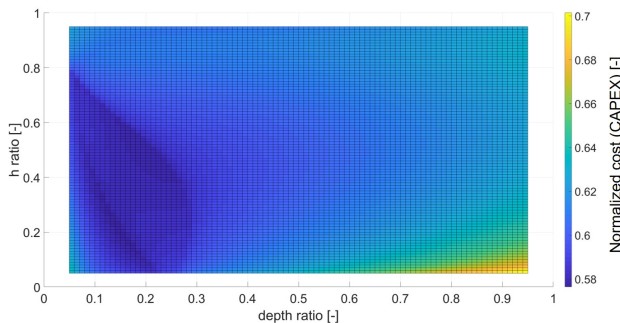

**Figure 4.** 2D view of the response surface, design no. 1, normalized cost (CAPEX)

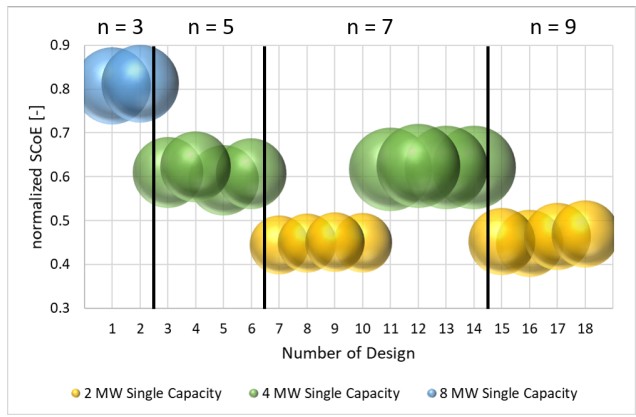

**Figure 5.** Normalized SCoE of the optimized SD-MRS designs

In Figure 5 the results of the optimized values for the simplified SCoE of the 18 SD-MRS designs are shown. The values are normalized to the value of a 20 MW SR. The bubble diameter indicates the overall SD-MRS capacity. With the exception of the Ennea rotors with three rows (Design no. 15 and no. 16), all designs with an increasing order of rotors are slightly favorable than the decreasing order. There, the increase in energy yield outweighs the higher tower base moment. The differences between designs no. 3 and no. 4 with designs no. 5 and no. 6 is due to a better load distribution, resulting from a different bar layout.

The three levels of SCoE values correlates with the used rotors in the designs. The designs with the lowest SCoE are the ones with the 2 MW rotors, on the intermediate SCoE level are the 4 MW rotors and on the highest SCoE level the 8 MW rotors.

This is due to the relative small fraction of tower and space frame mass on the overall mass and therefore cost. Design driver for the SD-MRS are the RNA masses, they benefit from smaller rotors based on cubic scaling.





# 4  MD-MRS

As a first venture into MD-MRS designs, the INNWIND design with 45 rotors is used in a slightly modified version. The number of rows and rotors per row are unchanged and the layout can be seen in Figure 6.

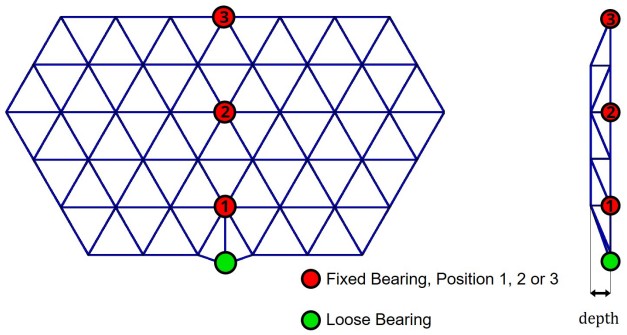

Figure 6. MD-MRS design for 45 rotors based on (Jamieson et al., 2017)

The overall capacity for the MD-MRS is set to the same values as the SD-MRS, to get a direct comparison. This results in the values shown in Table 3.

Table 3. Scaled rotor data for the 45 rotor MD-MRS

| $P_{overall}$ [MW] | $P_{rotor}$ [kW] | $D$ [m] | $m_{rotor}$ [t] | $m_{nacelle}$ [t] | $m_{RNA}$ [t] |
|---|---|---|---|---|---|
| 14 | 311.1 | 31.5 | 1.27 | 2.45 | 3.71 |
| 18 | 400.0 | 35.7 | 1.85 | 3.57 | 5.41 |
| 20 | 444.4 | 37.6 | 2.16 | 4.18 | 6.34 |
| 24 | 533.3 | 41.2 | 2.84 | 5.49 | 8.34 |
| 28 | 622.2 | 44.5 | 3.58 | 6.92 | 10.50 |

The design parameters to be varied are again the depth of the structure and the fixed bearing position and therefore the tower height. Instead of a quasi-continuous variation of the fixed bearing position over the height, three discrete positions are investigated. Position 1 in the second row at the bottom, position 2 in the middle of the space frame and position 3 at the top.

All three variants have the loose bearing in the first space frame row.

Figure 7 shows the normalized cost (CAPEX) of the 14 MW MD-MRS for the three fixed bearing positions plotted over the unit less total depth ratio. The cost are normalized to the cost value from the 20 MW SR. The total depth ratio indicates the ratio of depth to the width of the space frame. All three curves have a minimum of cost between 0.1 and 0.13 total depth ratio, meaning that the optimal design has a depth of 10%-13% of the space frame width. There the loads of the members have an optimal distribution, resulting in the lowest cost. Comparing the three fixed bearing positions, position 1 at the bottom




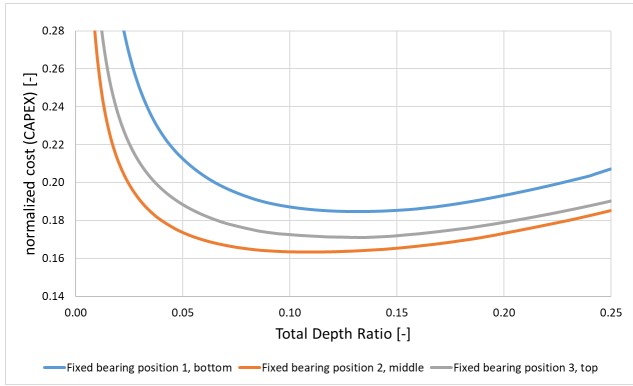

**Figure 7.** Normalized cost (Capex) of the 14 MW MD-MRS over design parameter changes

has universally the highest cost. At this position the space frame rests almost with the whole weight on the tower and on itself. Almost all members are in a compression state and need to be redimensioned due to stability after the initial strength dimensioning, resulting in high space frame cost. The upside of this fixed bearing position 1 is a relatively short tower compared to the other positions and therefore lower tower cost. For fixed bearing position 3 the behavior is quite the opposite. The highest
of the three tower versions is present, resulting in the highest tower cost. The space frame is hanging on the tower, resulting in tension state members without the need to redimension. This results in low space frame cost. Fixed bearing position 2 shows the overall lowest cost of the three fixed bearing positions. The MD-MRS designs with 18 MW to 28 MW are showing the same results: optimal total depth ratio of around 0.1 to 0.13 and an optimal fixed bearing position 2 in the middle of the space frame.

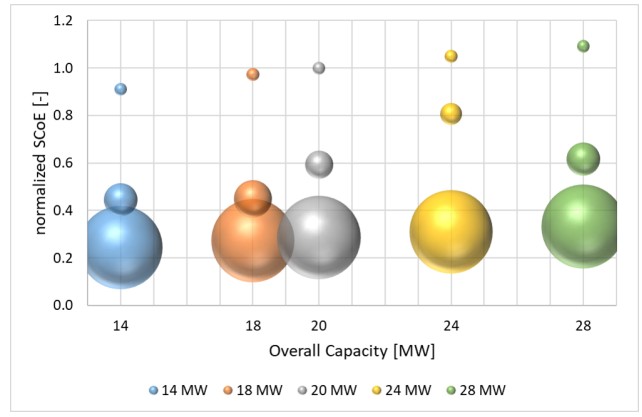

**Figure 8.** SD-MRS, MD-MRS and SR results for normalized SCoE

In Figure 8 the SCoE results of SD-MRS, MD-MRS and SR are presented, normalized to the 20 MW SR value. The size of the bubbles indicates the number of rotors, the smallest one indicate the SR, the intermediate sized bubbles label the SD-MRS

WIND
ENERGY
SCIENCE
DISCUSSIONS

and the big bubbles denote the MD-MRS with the fixed number of 45 rotors. The SD-MRS are presented by the one design with the lowest SCoE from each overall capacity level. The SR and the MD-MRS are progressing linear over the overall capacity, based on cubic scaling. The conclusion here would not be to build up a MRS with a high number of rotors and a small overall capacity. It just seems that way, since only a fixed number of rotors was investigated for the MD-MRS. The investigation with a variable, high number of rotors is missing and seems to be the next step. The Tri rotor shows the least potential to reduce cost, based on the relative large rotors and therefore cost.

## 5   Conclusions

The aim of this study was to develop a simplified method for preliminary calculations of masses and therefore cost for multi rotor wind turbine systems. The simplifications in the dimensioning process were used to avoid iterations for convergence and to allow a fast way to investigate a variety of designs and design parameters.

Several SD-MRS designs were designed, simulated and optimized regarding the cost. The space frame of a MRS is sensible to the design parameters, since the load distribution can change with the design. Members can change from a tension to the compression state or the other way around. Stability seems to have a big influence since many space frame members needed to be redimensioned, when in the compression state.

The SD-MRS designs with small single rotors showed the highest potential to reduce cost. One particular MD-MRS design with 45 rotors was also investigated and showed an optimal depth to width ratio for the space frame of 10%-13%. A fixed bearing position and therefore a tower height in the middle of the space frame was most promising. Overall the MD-MRS was showing more potential than the SD-MRS to reduce cost.

Next steps include more MD-MRS designs with variable number of rotors. After that, one or two promising designs will be analyzed in detail, regarding design load cases according to (IEC 61400-1 Ed.4), fatigue analysis, local shell/plate buckling and dynamic behavior (modal analysis).

*Author contributions.*  SSt performed all simulations, wrote the pre- and post-processing code, as well this paper. PDa and MTa helped formulate the ideas and gave technical advice in the regular discussions. PDa helped with the cost to mass factors. All reviewed this paper.

*Competing interests.*  The authors declare that they have no conflict of interest.

*Acknowledgements.*  The content of this paper was developed within the project X-Rotor/X-Multirotor, part of X-Energy, which is funded by the Federal Ministry of Education and Research (BMBF) and Siemens Gamesa Renewable Energy (SGRE), who we would like to thank at this place.



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
