# Peer review of "Simplified support structure design for multi rotor wind turbine systems"

_Wind Energy Science, 2020_

## Referee Comment (RC1) · Anonymous Referee #1 · 29 Mar 2020

The paper is well done and the content is good enough for a conference paper. As I understand the request the paper is for a conference. For a Journal paper the content is not enough and not for interest.

---

## Referee Comment (RC2) · Anonymous Referee #2 · 21 Apr 2020

The present is a clear, well structured, parametric analysis of the costs linked to the support structure for multi-rotor concepts, varying a number of key design parameters.

The need for this level of simplification is well justified, and the source of the hypotheses / assumptions are always cited, making this work a good step forward toward the investigation of multi-rotor wind turbines.

I did not identify any major technical error, the methodology is clear and well backed up with references. The paper is well written, with a logic structure and flow, and no typos/errors.

The only minor comment would be the following: it seems that the parametric investigation has been conducted with the aim of identifying an "optimum" or "optima", and

the identification of these optima has been done manually. Due to the simplifying assumption, this approach can be considered suitable.

Nonetheless, and especially considering a possible future work continuation, where the level of complexity of the problem will probably be enhanced, as well as the number of configurations, it would be suitable to frame the whole problem formally as an optimisation problem, and take advantage of state-of-the-art optimisation approaches.

---

## Author Response (AR1)

**Title: Simplified support structure design for multi rotor wind turbine systems**

**Author(s): Sven Störtenbecker et al.**

**MS No.: wes-2020-46**

**MS Type: Research article**

**Iteration: Revised Submission**

**Special Issue: Wind Energy Science Conference 2019**

**Anonymous Referee #1:**

(1) Comments from Referee:
The paper is well done and the content is good enough for a conference paper. As I understand the request the paper is for a conference. For a Journal paper the content is not enough and not for interest.

(2) Author's response:
Thank you very much for your comments.

The goal of this study was to introduce a simplified method for preliminary designs of support structures for multi rotor wind turbine systems. We think we achieved that and laid out the groundwork for subsequent, more detailed analyses.

(3) Author's changes in manuscript:
The simplified nature of this study is documented and no changes were being done.

**Anonymous Referee #2:**

(1) Comments from Referee:
The present is a clear, well structured, parametric analysis of the costs linked to the support structure for multi-rotor concepts, varying a number of key design parameters.

The need for this level of simplification is well justified, and the source of the hypotheses / assumptions are always cited, making this work a good step forward toward the investigation of multi-rotor wind turbines.

I did not identify any major technical error, the methodology is clear and well backed up with references. The paper is well written, with a logic structure and flow, and no typos/errors.

The only minor comment would be the following: it seems that the parametric investigation has been conducted with the aim of identifying an "optimum" or "optima", and the identification of these optima has been done manually. Due to the simplifying assumption, this approach can be considered suitable.

Nonetheless, and especially considering a possible future work continuation, where the level of complexity of the problem will probably be enhanced, as well as the number of

configurations, it would be suitable to frame the whole problem formally as an optimisation problem, and take advantage of state-of-the-art optimisation approaches.

(2) Author's response:
Thank you very much for your comments.

Yes, due to the assumptions and the simplified dimensioning procedure, the minimum for each design was found manually in this study.

You are right, with increasing complexity in future works, due to more design parameters and other criteria apart from material strength and stability, it would be more suitable to formulate the optimisation as a mathematical optimisation problem.

In the revised paper this will be added as a remark in the outlook.

(3) Author's changes in manuscript:
In line 194-196 (marked-up version) a remark was added to clarify the approach to find the minima in this study:
"These minima are determined manually with no incorporation into an overarching mathematical optimization approach of the dimensioning procedure."

In line 255-257 (marked-up version) a remark was added to indicate the possible need for a formal optimization approach:

[revised manuscript text omitted]